# Epidemiological evidence relating risk factors to chronic obstructive pulmonary disease in China: A systematic review and meta-analysis

Hong Chen[1], Xiang Liu[2], Xiang Gao[1], Yipeng Lv[1], Liang Zhou[1], Jianwei Shi[1], Wei Wei[3], Jiaoling Huang[1], Lijia Deng[4], Zhaoxin Wang[1], Ying Jin[3‡*], Wenya Yu[1‡*]

**1** School of Public Health, Shanghai Jiao Tong University School of Medicine, Shanghai, China, **2** Department of Respiratory Disease, The 903rd Hospital of PLA, Hangzhou, Zhejiang, China, **3** Department of general practice, Dapuqiao Community Health Service Center of Huangpu District, Shanghai, China, **4** School of informatics, The University of Leicester, Leicester, United Kingdom

☯ These authors contributed equally to this work.
‡ YJ and WY also contributed equally to this work.
* jsjyyuwenya@sina.cn (WY); jyhshf@126.com (YJ)

**Data Availability Statement:** All relevant data are within the paper and its Supporting Information files.

## Abstract

### Background

Chronic obstructive pulmonary disease (COPD), the most common chronic respiratory disease worldwide, not only leads to the decline of pulmonary function and quality of life consecutively, but also has become a major economic burden on individuals, families, and society in China. The purpose of this meta-analysis was to explore the risk factors for developing COPD in the Chinese population that resides in China and to provide a theoretical basis for the early prevention of COPD.

### Methods

A total of 2457 cross-sectional, case-control, and cohort studies published related to risk factors for COPD in China were searched. Based on the inclusion and exclusion criteria, 20 articles were selected. Stata 11.0 was used for meta-analysis. After merging the data, the pooled effect and 95% confidence intervals (CIs) were calculated to assess the association between risk factors and COPD. Heterogeneity between studies was assessed using $I^2$ and Cochran's $Q$ tests. Begg's test was used to assess publication bias.

### Results

Exposure to particulate matter less than 2.5 μm in diameter (PM2.5) (pooled effect = 1.73; 95%CI: 1.16~2.58; $P$ <0.01), smoking history (pooled effect = 2.58; 95%CI: 2.00~3.32; $P$ <0.01), passive smoking history (pooled effect = 1.39; 95%CI: 1.03~1.87; $P$ = 0.03), male sex(pooled effect = 1.70; 95%CI: 1.31~2.22; $P$ <0.01), body mass index (BMI) <18.5 kg/m² (pooled effect = 1.73; 95%CI: 1.32~2.25; $P$ <0.01), exposure to biomass burning emissions (pooled effect = 1.65; 95%CI: 1.32~2.06; $P$ <0.01), childhood respiratory infections (pooled effect = 3.44; 95%CI: 1.33~8.90; $P$ = 0.01), residence (pooled effect = 1.24; 95%CI:

**Funding:** This work was supported by grants from the Natural Science Foundation of China (71774116; 71603182), the Shanghai Excellent Young Talents Project in Health System (2018YQ52), the Shanghai Public Health Outstanding Young Personnel Training Program (GWV-10.2-XD07), the National Key R&D Program of China (2018YFC1314700), the Shanghai Pujiang Program (2019PJC072), and the Zhejiang Provincial Natural Science Foundation of China (LQ21H100001).The funders had no role in study design, data collection and analysis, decision to publish, or preparation of the manuscript.

**Competing interests:** The authors have declared that no competing interests exist.

1.09~1.42; $P$<0.01), and a family history of respiratory diseases (pooled effect = 2.04; 95% CI: 1.53~2.71; $P$<0.01) were risk factors for COPD in the Chinese population.

## Conclusion

Early prevention of COPD could be accomplished by quitting smoking, reducing exposure to air pollutants and biomass burning emissions, maintaining body mass index between 18.5 kg/m$^2$ and 28 kg/m$^2$, protecting children from respiratory infections, adopting active treatments to children with respiratory diseases, and conducting regular screening for those with family history of respiratory diseases.

## Introduction

Data from the World Health Organization show that chronic obstructive pulmonary disease (COPD) has become an important contributor to the global burden of non-communicable diseases [1]. From 1990 to 2017, the prevalence of COPD showed an overall upward trend with a relative increase of 5.9%. In 2017, the global prevalence rate of COPD was approximately 3.92%. COPD is also the most common cause of death in patients with chronic respiratory diseases. Data show that in 2017, an average of 41.9 people died of COPD per 100,000 people, accounting for 5.7% of all deaths. In China, COPD has become the third most common chronic disease, with the prevalence of 4.71% in 2017 [2] and the mortality rate of 0.068% [3]. In addition, there were specific characteristics of the development of COPD in Chinese population compared with other groups due to the impact of climate change, environmental pollution, public health literacy and medical technology. Furthermore, the incidence rate of COPD was estimated to be more severe in the future. However, the prevention and control of COPD in China is far from enough.

The main clinical symptoms of COPD that greatly affect the quality of life are chronic cough, sputum expectoration, and shortness of breath after physical activity [4]. Complications such as osteoporosis [2], a decreased ability to keep balanced [5], cardiovascular diseases [6], dysphagia [7], and depression [8] are common in patients with COPD, which further increase the number of acute exacerbations, hospitalization rate, and mortality of patients with COPD and seriously affect the prognosis and quality of life of patients. In addition, patients with COPD generally have a long course of disease, and the condition continues to deteriorate over time. Because patients with advanced COPD have a decreased ability for self-care in daily life and increased disability, their family caregivers have assumed a huge financial burden [9] and experience mental stress [10].

The occurrence of COPD is not only driven by genetic factors but also by environmental factors and demographic characteristics. In domestic studies [11–13], factors such as exposure to smoke (smoking, air pollution, occupational dust, and chemicals), residential radon, inhaled corticosteroids, a low body mass index (BMI), age, sex, socioeconomic status, lung hypoplasia, asthma, airway hyper-responsiveness, HIV infection, and genetic polymorphisms were associated with the occurrence and development of COPD. A cross-sectional study conducted by Chen [14] in 10 provinces in mainland China found that smoking, environmental air pollution, underweight, chronic coughing in children, a history of parental respiratory diseases, and low education levels were the main risk factors for COPD in the Chinese population. A meta-analysis by Yang [15] pointed out that male sex, smoking, low education level, low BMI (<18.5 kg/m$^2$), family history of respiratory diseases, history of allergies, childhood respiratory

infections, repeated respiratory infections, exposure to occupational dust and biomass burning emissions, poor residential ventilation, and living in and around polluted areas may be important risk factors for COPD in mainland China. Foreign studies [16–19] also found that altitude, periodontal pathogens, and the intake of processed and unprocessed red meat were significantly correlated with COPD. Research by Busch [20] showed that genes associated with lung function play a role in a person's susceptibility to COPD. However, there are still many limitations to the existing studies because the occurrence of COPD is associated with environmental, genetic, and other factors. Most of the current research on COPD in China is still based on cross-sectional, case-control studies and other research types with a weak form of evidence. Prospective studies, especially large population cohorts were less frequently conducted due to the difficulty of implementation; the diagnostic criteria, measurements of exposure, and distribution of sample characteristics in different studies are not all the same. Thus, horizontal comparison is difficult. In contrast, foreign researches focus on various race groups, and the results and conclusions of these studies have limited relevance in the early prevention of COPD in the Chinese population. In addition, existing meta-analyses often include retrospective observational studies alone and lack cohort studies with stronger, more reliable causal links. Further, the research included in the meta-analyses are mostly of a single area; thus, the results of the study are not representative.

This study aimed to conduct a meta-analysis on populations in multiple regions of China and to integrate various studies (including cohort, case-control, and cross-sectional studies) to explore potential risk factors for COPD in Chinese residents. This study also hopes to provide a theoretical basis for the early identification and prevention of high-risk COPD.

## Methods

The Preferred Reporting Items for Systematic Review and Meta-Analyses (PRISMA) statement was employed to design and report the study. All studies designed to describe risk factors for COPD were searched.

### Search strategy

English and Chinese databases such as Web of Science, PubMed, CNKI and WanFang were searched using MESH terms: "COPD," "Chronic Obstructive Pulmonary Disease," "risk factors," "case-control study," "cross-sectional study," and "cohort study." Literature tracing and manual retrieval were also used to collect relevant literature published from January 1, 2000 to November 1, 2021. Articles associated with risk factors for COPD were initially screened using "COPD" and "risk factors," and then retrieved from the preliminary screening results using "case-control study," "cross-sectional study," and "cohort study." (S1 Table)

### Study selection

Inclusion criteria were as follows. (1) Publicly published case-control, cohort, or cross-sectional studies on the risk factors for COPD. (2) Study population: The objects of all studies refers to Asian population that always live in China. (3) The definition of exposure is similar; for example, BMI $<18.5$ kg/m$^2$ indicates underweight and $\geq 28$ kg/m$^2$ indicates obesity. (4) The case diagnosis is clear and was confirmed clinically. We defined COPD patients as subjects with FEV1 / FVC less than 70% after using post-bronchodilator, or diagnosed with chronic bronchitis, emphysema or other diseases dominated by airflow restriction by doctors.(5) The research results in the article provide the odds ratio (OR), risk ratio (RR), or at least the basic data for OR/RR calculations. Exclusion criteria were as follows. (1) repeated research. (2) OR/ RR and 95% confidence intervals (CIs) were not provided and could not be calculated. (3) No

confounders were adjusted. The initial search and selection of literature were completed by two authors (H Chen and X Liu) independently. Literature were screened according to the title and abstract, and those not meeting the inclusion criteria were excluded.

### Data extraction

The data were extracted by two independent reviewers (X Gao and YP Lv), and judged by another author when contradictions occur. All selected data were arranged as a standard data, including: (I) the first author; (II) year of publication; (III) the area of the research; (IV) sample size; (v) type of research method; (vi) OR/RR values and 95%CIs for potential risk factors.

The quality of the cross-sectional studies was evaluated according to the Agency for Healthcare Research and Quality Literature Quality Evaluation Scale [21], including 11 standards. The quality of case-control and cohort studies were evaluated according to the Newcastle-Ottawa Scale (NOS) [22], including eight standards. Each standard score of the above literature quality evaluation scale was different. Each standard score of the above literature quality evaluation scale was different. The full score was 10 points; Studies that get $\geq 8$ points were considered to be of high quality, 5–7 points matched the criteria of medium-quality studies and $<5$ points were considered to be poor-quality studies. Two researchers (L Zhou and JW Shi) independently evaluated the quality of each included study. (S1 File)

### Statistical analyses

Statistical analysis was performed using Stata 11.0 (StataCorp, College Station, TX, USA). The results were reported as pooled effect with the corresponding 95%CI, and $P < 0.05$ was considered significant. Cochran's $Q$ and $I^2$ tests were used to evaluate the heterogeneity of the included studies. $I^2$ which ranges from 0 to 100% denotes the percentage of the variability in effect estimates that is due to heterogeneity rather than sampling error (chance). We used the random-effects model when heterogeneity across the studies was large ($I^2 > 50\%$, $P < 0.05$) and fixed-effects meta-analysis at small heterogeneity ($I^2 < 50\%$, $P < 0.05$) [23]. When large heterogeneity was present, sensitivity analysis and subgroup analysis were performed to identify responsible outlier studies. The Begg's test was used to evaluate the publication bias of the included studies, and $P < 0.10$ was considered statistically significant.

## Results

### Study selection

A total of 2457 articles were identified using both database and manual searches (Fig 1). After duplicate researches were excluded, we then excluded 1672 articles based on titles and abstracts. comments and articles. After reading the full text of the remaining 335 articles, 315 papers were eliminated due to inaccurate exposure definitions or outcome diagnoses, incomplete data and unadjusted confounding factors. Finally, 20 articles with 995190 participants were selected and included in this study. (Table 1)

### Study descriptions

All the included studies were performed in China. Five studies were published in Chinese [25, 28, 31, 32, 34], and the others were published in English. Three studies were conducted as case-control studies, five studies were conducted as cohort studies, and the others were conducted as cross-sectional studies.

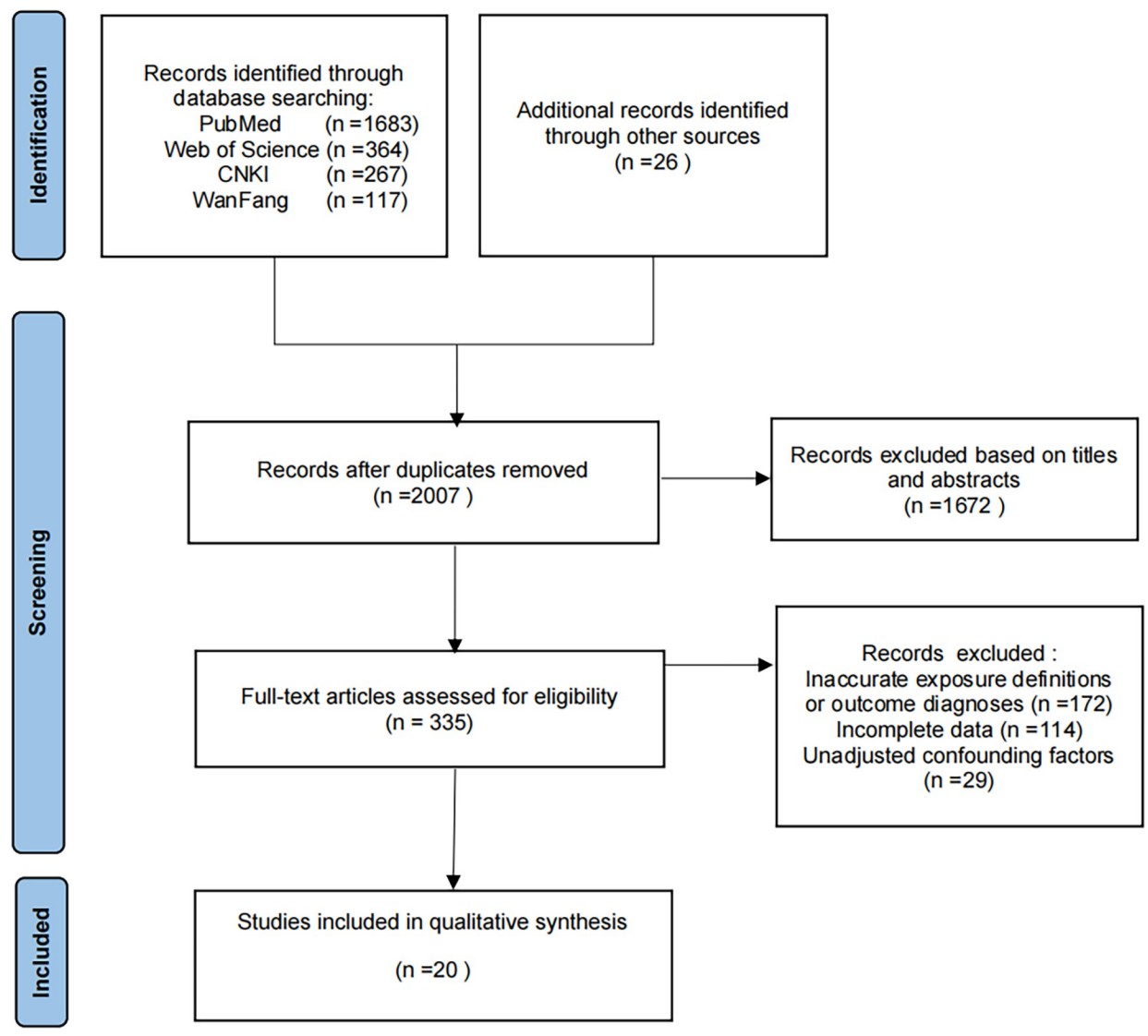

**Fig 1. PRISMA flow diagram for selection of studies.**

## Assessment of heterogeneity

Clinical and methodological diversity between studies led to statistical heterogeneity. The results of heterogeneity test were listed in Table 2. (Table 2) Heterogeneity was found among the potential risk factors of exposure to particulate matter less than 2.5 μm in diameter (PM2.5), smoking history, passive smoking history, sex, BMI $\geq$28 kg/m$^2$, BMI <18.5 kg/m$^2$, exposure to biomass burning emissions,family history of respiratory diseases and childhood respiratory infections. Therefore, we calculated the pooled effect values of these factors using a random-effects model.

## Risk factors

Meta-analysis results showed that exposure to PM2.5 (pooled effect = 1.73; 95%CI: 1.16~2.58; $P$ <0.01; $I^2$ = 65.7%), smoking history (pooled effect = 2.58; 95%CI: 2.00~3.32; $P$ <0.01; $I^2$ = 78.5%), passive smoking history (pooled effect = 1.39; 95%CI:1.03~1.87; P = 0.03; $I^2$ = 59.5%),

**Table 1. Characteristics of included trials and methodological quality assessments.**

| First author | Year of publication | Study region | Sample size | Study design | Score |
|---|---|---|---|---|---|
| XP Yan [24] | 2020 | Suzhou | 4725 | cross-sectional study | 5 (middle) |
| C Wang [14] | 2018 | China | 50991 | cross-sectional study | 9 (high) |
| YM Tang [25] | 2018 | Hubei | 2389 | cross-sectional study | 5 (middle) |
| YT Peng [26] | 2018 | Hunan | 638 | cross-sectional study | 7 (middle) |
| CJ Zhao [27] | 2018 | Haikou | 9432 | cross-sectional study | 5 (middle) |
| YE Zhang [28] | 2018 | Ningxia | 1800 | cross-sectional study | 5 (middle) |
| S Liu [29] | 2017 | Guangdong | 5993 | cross-sectional study | 9 (high) |
| YP Ding [30] | 2015 | Hannan | 5463 | cross-sectional study | 6 (middle) |
| JH Yu [31] | 2015 | Chongqing | 3000 | cross-sectional study | 5 (middle) |
| G Hou [32] | 2012 | Shenyang | 2194 | cross-sectional study | 5 (middle) |
| NS Zhong [33] | 2007 | China | 20245 | cross-sectional study | 7 (middle) |
| PX Ran [34] | 2006 | China | 5111 | cross-sectional study | 6 (middle) |
| F Xu [35] | 2005 | Nanjing | 29319 | cross-sectional study | 8 (high) |
| JC Li [36] | 2020 | China | 452259 | cohort study | 10 (high) |
| JC Li [37] | 2019 | China | 393444 | cohort study | 10 (high) |
| YM Zhou [38] | 2013 | Guangzhou | 2577 | cohort study | 8 (high) |
| P Yin [39] | 2007 | Guangzhou | 891 | cohort study | 6 (middle) |
| HC Huang [40] | 2019 | Taiwan | 3941 | case-control study | 8 (high) |
| TC Chan [41] | 2015 | Taiwan | 200 | case-control study | 9 (high) |
| M Chan-Yeung [42] | 2007 | Hong Kong | 578 | case-control study | 7 (middle) |

male sex (pooled effect = 1.70; 95%CI: 1.31~2.22; $P$ <0.01; $I^2$ = 87.1%), BMI <18.5 kg/m$^2$ (pooled effect = 1.73; 95%CI: 1.32~2.25; $P$ <0.01; $I^2$ = 93.5%), exposure to biomass burning emissions (pooled effect = 1.65; 95%CI: 1.32~2.06; $P$ <0.01; $I^2$ = 88.0%), childhood respiratory infections (pooled effect = 3.44; 95%CI: 1.33~8.90; $P$ = 0.01; $I^2$ = 96.6%), residence (pooled effect = 1.24; 95%CI: 1.09~1.42; $P$ <0.01; $I^2$ = 0.00%) and family history of respiratory diseases (pooled effect = 2.04; 95%CI: 1.53~2.71; $P$ <0.01; $I^2$ = 88.6%) had a significant impact on the Chinese population's risk of developing COPD. Drinking history (pooled effect = 0.82; 95%CI:

**Table 2. Results of meta-analysis and heterogeneity test.**

| Risk factors | Number of studies | Meta-analysis | | Heterogeneity | | Meta analytical model |
|---|---|---|---|---|---|---|
| | | Pooled effect (95%CI[b]) | P-value | $I^2$ (%) | P-value | |
| Exposure to PM2.5 | 3 | 1.73(1.16~2.58) | <0.01 | 65.7% | 0.05 | Random |
| Smoking history | 12 | 2.58(2.00~3.32) | <0.01 | 78.5% | <0.01 | Random |
| Passive smoking history | 4 | 1.39(1.03~1.87) | 0.03 | 59.5% | 0.06 | Random |
| Drinking history | 2 | 0.82(0.54~1.23) | 0.37 | 0.0% | 0.75 | Fixed |
| Male sex | 7 | 1.70(1.31~2.22) | <0.01 | 87.1% | <0.01 | Random |
| BMI [a] <18.5 kg/m$^2$ | 10 | 1.73(1.32~2.25) | <0.01 | 93.5% | <0.01 | Random |
| BMI ≥28 kg/m$^2$ | 8 | 0.96(0.76~1.22) | 0.75 | 75.9% | 0.01 | Random |
| Exposure to biomass burning emissions | 7 | 1.65(1.32~2.06) | <0.01 | 88.0% | <0.01 | Random |
| Childhood respiratory infections | 4 | 3.44(1.33~8.90) | 0.01 | 96.6% | <0.01 | Random |
| Residence | 5 | 1.24(1.09~1.42) | <0.01 | 0.0% | 0.96 | Fixed |
| Family history of respiratory diseases | 5 | 2.04(1.53~2.71) | <0.01 | 88.6% | <0.01 | Random |

[a] BMI, Body mass index

[b] CI, 95% confidence intervals

0.54~1.23; $P$ = 0.37; $I^2$ = 0.00%) and body mass index (BMI) ≥28 kg/m$^2$ (pooled effect = 0.96; 95%CI: 0.76~1.22; $P$ = 0.75; $I^2$ = 75.9%) are not associated with COPD of Chinese population. (Table 2)

## Sensitivity analysis

Sensitivity analysis was performed to evaluate the stability and reliability of the results. In our study, there was no significant difference in the pooled effect before and after excluding study with high heterogeneity or low quality, which indicated that the results of sensitivity analysis are reliable. (Fig 2)

## Publication bias

The Begg's test was used to assess potential publication bias. The results of the Begg's test showed that there was a certain degree of asymmetry in the scatter points corresponding to exposure to biomass burning emissions (Fig 3), therefore the trim and fill analysis [43] was further performed and showed no further studies required. The other risk factors did not have significant publication bias ($P$>0.10). (S2 Table)

## Subgroup analysis

In order to address potential confounding and reduce heterogeneity, we performed several subgroup analyses by the source of research object (hospital, population), research method (case-control study, cross-sectional study, cohort study), geographic region (national, single province) and research duration (<5 years, ≥5 years). Stratifying our analysis resulted in a reduction of heterogeneity, which still exists. (S1 Fig)

## Discussion

This meta-analysis showed that exposure to PM2.5, smoking history, passive smoking history, BMI <18.5 kg/m$^2$, exposure to biomass burning emissions, childhood respiratory infections and family history of respiratory diseases were risk factors for COPD in the Chinese population.

Air pollution suspended in moist air is usually called "smoke," which comprises dust particles of different sizes, non-metal oxides, organic compounds, and heavy metals [44]. Harmful substances in smoke can cause bronchospasms that increase airway resistance. Long-term exposure to smoke can lead to the occurrence of COPD [45]. This is consistent with the discovery of Mark *et al*. that an increase in exposure to smoke over a lifetime can lead to a significant increase in the risk of COPD [46]. As a risk factor for COPD in the Chinese population, exposure to smoke mainly includes exposure to PM2.5, smoking, passive smoking, and exposure to biomass burning emissions.

Atmospheric particulate matter pollution is an important factor affecting the course of various respiratory and cardiovascular diseases and is associated with a higher risk of cardiopulmonary mortality and morbidity [47]. Increasing evidence shows that PM2.5 is the most harmful air pollutant to human health. Long-term exposure to PM2.5 can induce a decline in lung function, emphysema, and changes to airway inflammation [48]. Animal experiments have shown that PM2.5 promotes lung inflammation and oxidative stress in mice [49]. Further, excessive inflammation and oxidative stress cause or aggravate respiratory diseases. A study in France showed that high exposure to PM2.5 was significantly associated with a decrease in serum cytokine levels. PM2.5 induces the expression of inflammatory cytokines in human bronchial epithelial cells through multiple pathways [50]. This change in cytokine

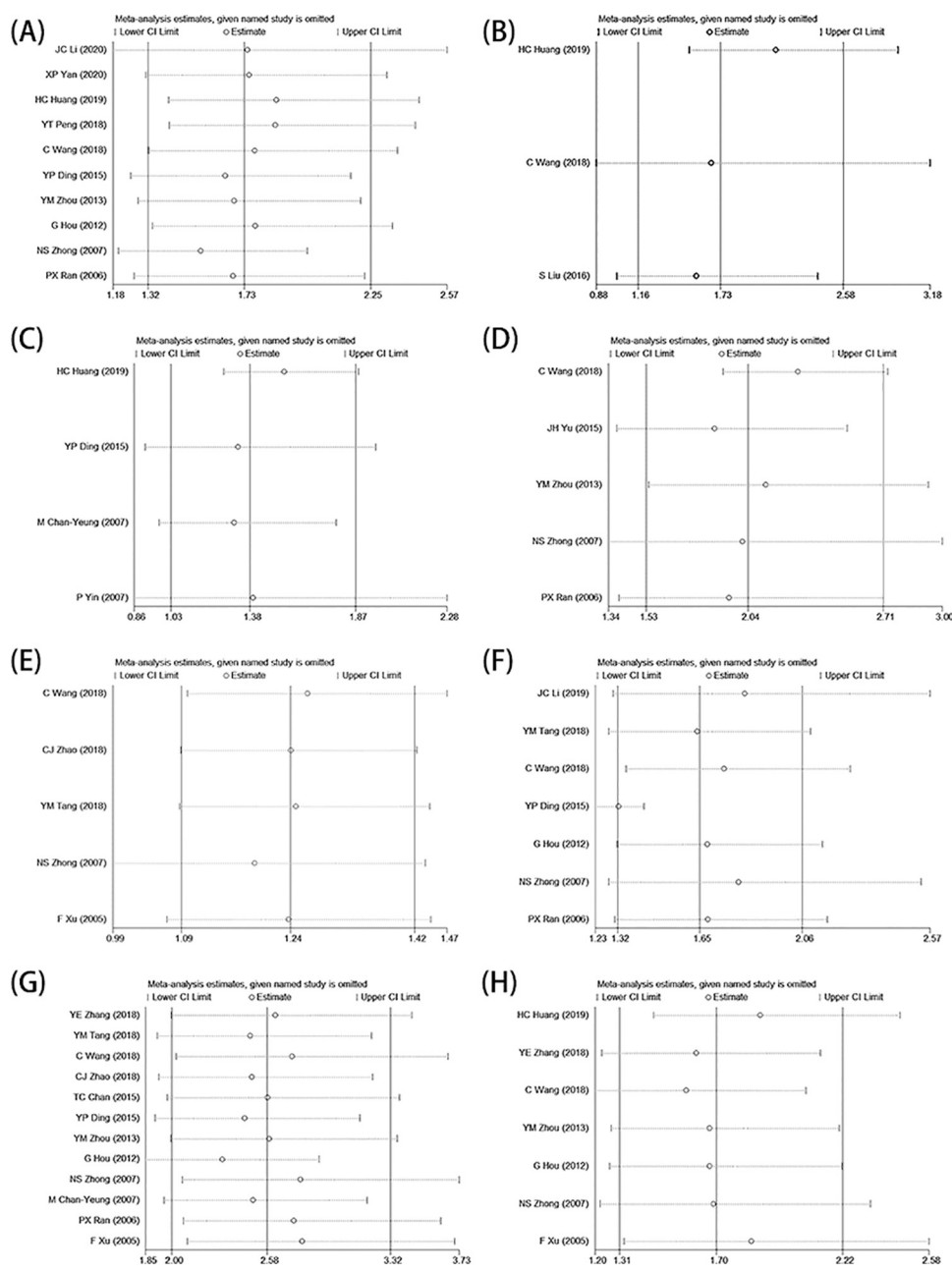

**Fig 2. Results of sensitivity analysis.** (A): BMI <18.5 kg/m$^2$. (B): Exposure to PM2.5. (C): Passive smoking history. (D): Family history of respiratory diseases. (E): Residence. (F): Exposure to biomass burning emissions. (G): Smoking history. (H): Male sex.

levels can become one of the main causes of COPD by disrupting the balance of the immune response [51].

Smoking is the most important risk factors for COPD. A previous meta-analysis showed that the incidence of COPD among ex-smokers and current smokers was higher than that among never-smokers (*RR* values were 2.35, 2.89, and 3.51, respectively) [52]. Active or passive inhalation of cigarette smoke by the human body can cause damage to the respiratory mucosa, leading to chronic inflammation of the respiratory tract [53]. Animal experiments have shown

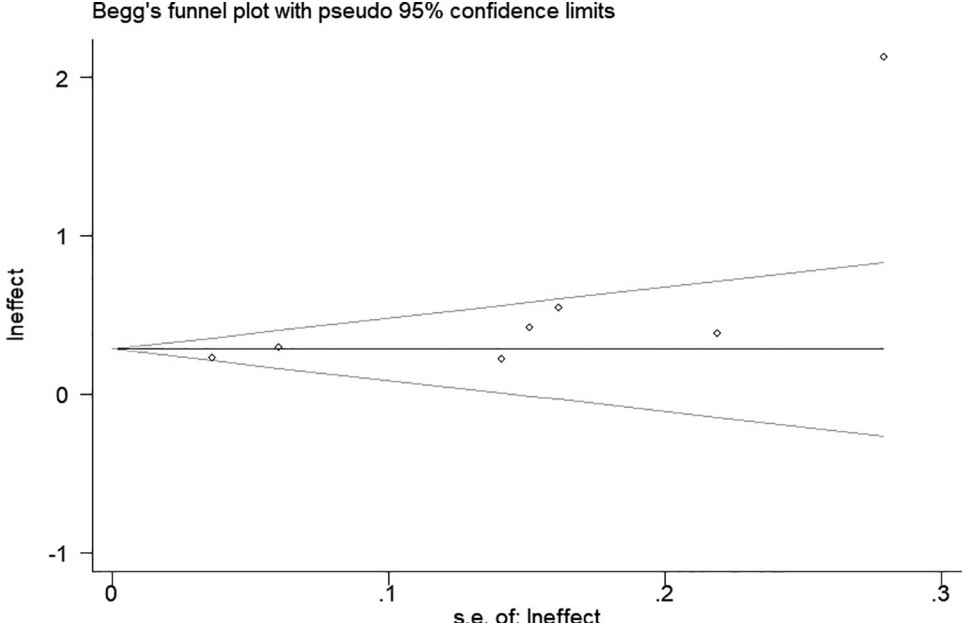

**Fig 3. Begg's test.** Exposure to biomass burning emissions.

that smoking can promote the occurrence and development of COPD through a variety of mechanisms, including hypersecretion of airway mucus, increased inflammatory cells in the airway lumen and lung parenchyma such as neutrophils and macrophages, thickening of the airway wall of lung tissue, and excessive deposition of collagen-based extracellular matrix [54]. Poorly ventilated households that use biomass fuels, including wood, animal manure, and crop residues, for cooking and heating in developing countries [55] and women and children have the highest rate of exposure to biomass burning emissions [56].

Organic and inorganic compounds and insoluble particles produced by burning biomass play an important role in inflammatory reactions, which can adversely affect the lung parenchyma, interstitium, and vasculature, thereby affecting the occurrence and development of COPD. Many people, especially in economically underdeveloped countries, cook and heat through using open fire, fuel, coal and simple stoves to burn biomass such as wood, animal manure and crop waste. It is easy to cause airway obstruction and sustained lung damage if long-term exposure to biomass, which results in an increased risk of COPD. A meta-analysis conducted by a Chinese scholar [57] showed that biomass smoke exposure was a risk factor for COPD among Chinese residents. Meanwhile, another scholar [58] found that the exposure of biomass smoke was positively correlated with the risk of developing COPD.

Notably, our results suggest that male sex may be a potential risk factor for COPD. Some studies suggested men are more likely to develop COPD. This may be associated with higher exposure to tobacco in men and changes in sex hormones in women after menopause [59]. However, some studies have shown that the prevalence of COPD has increased faster in women than it has in men in recent years [60]. This may be due to the narrower inner diameter and higher sensitivity of airways [61], more susceptible to risk factors such as biofuels and air pollution, and weaker immune regulation and stronger inflammatory responses in females than that in males. Sex hormones have complex effects on the production of COPD. Matteis *et al*. indicated that sex hormones in the menstrual cycle affect bronchial responsiveness and PC20FEV1.0 decrease during the follicular phase of the menstrual cycle in about 30% of

women [62]. In addition, Firas *et al.* found that gender significantly influences the levels of inflammatory cytokines in female patients with COPD, and correlates with different clinical and physiological parameters [59].

Low body weight (BMI <18.5 kg/m$^2$) is a risk factor for COPD. Compared with a BMI in the normal range, a low BMI is associated with a faster decline in the forced expiratory volume in 1s [63]. A study by Rabinovich *et al.* showed that a decrease in BMI had a negative impact on the clinical outcomes of patients with COPD [64]. A decrease in BMI causes atrophy and a decrease in the strength of respiratory muscles. This leads to a decrease in the vitality of lymphocytes and macrophages and in the production of immunoglobulin and complement, which increases the likelihood of respiratory infections and inflammation [65].

To prevent COPD, it is important to remain vigilant on matters regarding the health of the respiratory system. Lung growth and development is affected by exposure during pregnancy, birth, childhood and adolescence, and any factors affecting lung growth and development during pregnancy and childhood may increase the risk of COPD. A study in the United Kingdom showed that childhood respiratory infections had long-term adverse effects on the lungs, including frustration of the respiratory tract, impaired development of lung parenchyma, and lung growth disorders, which may lead to COPD in adulthood [66]. As a result, we recommend taking interventions to protect children from respiratory infections or adopting active treatments to children with respiratory tract infection for early prevention of COPD. In addition, because the lungs of children aged 0–18 years are immature and still undergoing growth, we believe that more attention should be paid to the pulmonary infection among children aged 0–18 years.

In addition, a large number of studies have shown that the incidence of COPD is not only associated with the aforementioned environment, living habits, and other acquired factors, but is also affected by genetic factors. To date, multiple genomic regions have been found to be associated with the COPD phenotype. McCloskey *et al.* [67]suggested that genetic determinants may interact with smoke to affect susceptibility to COPD. However, no genetic markers were found in the included studies. In addition, our study showed that family history of respiratory diseases (patients whose parents and / or siblings have one of chronic bronchitis, emphysema, COPD, and bronchial asthma were counted as those with family history) was also a risk factor for COPD. Studies have confirmed that there is family aggregation in COPD, however, it is difficult to distinguish whether the family aggregation of patients is caused by genetic factors or environmental factors. Therefore, more research is needed for analysis.

This meta-analysis had certain limitations. First, the most noteworthy limitation of this study was the existence of a large number of heterogeneity. In the included studies, part of the studies having only a single area included, and some studies having varying areas included. Subgroup analysis results shown that the $I^2$ of each group is less than that of the whole group after stratification according to the geographic region (national, single province), which indicated that region is one of the reasons for high heterogeneity. In addition, almost a quarter of the included studies in this meta-analysis were hospital-based, which may have introduced bias. Second, although many of the included studies involved age as a factor, we did not analyze age as a potential risk factor because of the large inconsistencies in age division. Third, some studies mentioned that asthma and occupational exposure may also be potential risk factors for COPD, although due to the small amount or low quality of relevant literature, no specific analysis was performed in this meta-analysis on these factors. Fourth, we tried to explore the relationship between long-term exposure to PM2.5 and disease, however, the specific year of exposure was not indicated in the included literatures, which was a limitation of the study. Last, the literature included in this study did not explain the age, type of infection (virus / bacteria / fungi) and severity of infection in children with respiratory tract infection. Therefore, it

is too vague to state that any respiratory infection during childhood could lead to COPD. Therefore, it is necessary to further study the effects of age, asthma, occupational exposure, long-term exposure to PM2.5 and respiratory tract infections of different types and degrees in childhood on the occurrence of COPD in the Chinese population.

## Conclusion

Factors related with smoking exposure, body weight, and respiratory infections were identified as significant risk factors and potential preventive strategies for COPD. For the early prevention of COPD, clinicians and public health experts should advocate smokers to quit smoking and never-smokers not to start smoking. The government authorities should take into serious considerations for the measures to reduce air pollution and biomass burning emissions. Body mass index should be encouraged for everyone to be maintained between 18.5 kg/m$^2$ and 28 kg/m$^2$. Child health providers should take interventions to protect children from respiratory infections or adopt active treatments to children with respiratory infections. A regular screening is of great significance for people with family history of respiratory diseases.

## Supporting information

**S1 Checklist. PRISMA 2020 checklist.**
(DOCX)

**S1 Fig. Subgroup analysis.**
(TIF)

**S1 Table. Search strategy.**
(DOCX)

**S2 Table. Publication bias associated with potential risk factors for COPD.**
(DOCX)

**S1 File. Quality evaluation of included studies.**
(XLSX)

**S1 Data.**
(XLSX)

## Author Contributions

**Conceptualization:** Hong Chen, Yipeng Lv, Liang Zhou, Jianwei Shi.

**Data curation:** Hong Chen, Wei Wei, Jiaoling Huang.

**Formal analysis:** Hong Chen, Lijia Deng.

**Methodology:** Xiang Liu, Xiang Gao.

**Supervision:** Zhaoxin Wang, Ying Jin, Wenya Yu.

**Writing – original draft:** Hong Chen, Xiang Liu, Xiang Gao.

**Writing – review & editing:** Yipeng Lv, Liang Zhou, Jianwei Shi, Wei Wei, Jiaoling Huang, Lijia Deng, Zhaoxin Wang, Ying Jin, Wenya Yu.

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
