## [Decision Letter · Decision Letter 0]

27 Oct 2021

PONE-D-21-16397

Epidemiological evidence relating risk factors to Chronic Obstructive Pulmonary Disease in China: A systematic review and meta-analysis.

PLOS ONE

Dear Dr. Wenya Yu,

Thank you for submitting your manuscript to PLOS ONE. After careful consideration, we feel that it has merit but does not fully meet PLOS ONE’s publication criteria as it currently stands. Therefore, we invite you to submit a revised version of the manuscript that addresses the points raised during the review process.

We look forward to receiving your revised manuscript.

Kind regards,

Surasak Saokaew, PharmD, PhD, BPHCP, FACP

Academic Editor

PLOS ONE

Journal Requirements:

2. Please confirm that you have included all items recommended in the PRISMA checklist including:

- the full electronic search strategy used to identify studies with all search terms and limits for at least one database.

- a Supplemental file of the results of the individual components of the quality assessment, not just the overall score, for each study included.

- See https://journals.plos.org/plosmedicine/article?id=10.1371/journal.pmed.1000100#pmed-1000100-t003 for guidance on reporting.

4. Please upload a new copy of Figures 1,2,3 and 4 as the detail is not clear. Please follow the link for more information: " ext-link-type="uri" xlink:type="simple">https://blogs.plos.org/plos/2019/06/looking-good-tips-for-creating-your-plos-figures-graphics/"
" ext-link-type="uri" xlink:type="simple">https://blogs.plos.org/plos/2019/06/looking-good-tips-for-creating-your-plos-figures-graphics/"

Reviewers' comments:

Reviewer's Responses to Questions

**Comments to the Author**

1. Is the manuscript technically sound, and do the data support the conclusions?

Reviewer #1: Partly

Reviewer #2: Partly

Reviewer #3: Partly

2. Has the statistical analysis been performed appropriately and rigorously? 

Reviewer #1: No

Reviewer #2: Yes

Reviewer #3: Yes

3. Have the authors made all data underlying the findings in their manuscript fully available?

Reviewer #1: No

Reviewer #2: Yes

Reviewer #3: Yes

4. Is the manuscript presented in an intelligible fashion and written in standard English?

Reviewer #1: No

Reviewer #2: Yes

Reviewer #3: Yes

5. Review Comments to the Author

Reviewer #1: Review comments

In this study, the authors explored epidemiological evidence regarding risk factors for chronic obstructive pulmonary disease in the Chinese population using systematic review and meta-analytic methods. Although the titles and the research question were interesting, several serious methodologic concerns must be addressed before being re-considered for publication. I suggest that this manuscript undergo a major revision.

Introduction

• The structure of the introduction section should be re-organized. It is still not clear to me why this meta-analysis should be conducted. More detail on the study rationale should be added. Suppose the authors were to conduct a study to explore COPD risk factors specifically for the Chinese population. In that case, there should be a solid theoretical background of why there is a good reason to believe that COPD risk factors in the general population should be different from the Chinese population.

Methods

It is questionable whether restricting comprehensive literature searches to only English biomedical databases was adequate for a rather specific question for a specific population. Shouldn’t large, high-quality studies reported in Chinese be included?

A searching strategy (i.e., keywords used, number of records identified for each keyword) for each database should be provided in a supplementary appendix.

The study selection criteria were unclear and subjective. Please clarify all of the criteria and make it more objective. For example, how do you define a strictly controlled research method?, how do you define when the study population is clearly defined?, how do you define when the case diagnosis is clear? Etc.

Please change the terms “relative risk” to “risk ratio”.

“Literature review type articles” should not be stated in the exclusion criteria as the inclusion criteria had already stated that only published case-control, cohort, or cross-sectional studies would be included.

I am not quite sure that research with inaccurate design methods, low reliability, or poor quality should be excluded. I believe the idea of performing a systematic review was to comprehensive include all relevant studies possible. Exclusion of studies with poor methodological design might give rise to publication bias. Including these studies within this systematic review and performing a subgroup or sensitivity analyses to exclude these problematic studies might be a good alternative way.

Details on who perform the screening, searching, and selection of records should be included within the methods section.

In the data extraction section, “other basic information” should be further clarified. Also, within this section, potential factors that the authors intended to explore should be pre-specified.

In the statistical analyses section, pooling methods should be specified. The authors should pre-specified in detail when to perform fixed effects pooling or random effects pooling.

Concerning I-squared statistics, the authors should specify the cut-off used for significant heterogeneity (and reference should be provided).

Again, I’m still not convinced that only studies with moderate-to-high quality should be included in this meta-analysis. I believe that this would lead to publication bias.

Results

The authors stated in the results section that “257 papers were eliminated due to incomplete data, no adjustment for confounding factors, or results that were not significant.” However, these elimination criteria were not specified within the methods section.

Again, I’m not convinced that excluding studies with insignificant results is a good approach for conducting systematic review.

The sample size was reported as number of COPD cases, or the number of total people included? This should be clarified or separately reported.

It is not clear how the authors arrange the sequence of the included studies on Table 1. I suggest that the authors stratify this table by study design and arrange the studies within each category by year of publication.

Details on how the authors define considerable heterogeneity should be stated in the methods section, not in the results section.

Table 2, Table 3, and Figure 2 should be combined.

In Figure 2, not all significant predictors were included. It was unclear why the authors chose only some predictors to be presented in this figure.

In Table 2 and Table 4, “=” should be changed to “≥”.

Table 4 should be included as supplementary material.

In the results section, the authors did not provide any numerical, tables, figures for subgroup analyses.

Overall, the results section should be re-written. Essential data should be presented.

Conclusions

• The study conclusion is wrong. This meta-analysis did not show that reducing the exposure to risk factors would prevent or reduce the incidence of COPD. This meta-analysis only explores potential risk factors for COPD in the Chinese population.

Figures

• PRISMA flow should be updated to the 2020 version.

• The quality of all figures should be improved.

Reviewer #2: General comments:

The authors demonstrated the risk factors of COPD performing the systematic review and meta-analysis in Chinese population. Their article is likely to help readers to learn this field. According to their results, PM2.5, smoking history, BMI18.5kg/m2, exposure to biomass burning emissions, and family history of respiratory diseases were the risk factors. These findings let readers reconsider this field. Despite no description of new insights in this field, the review for each section has been adequately addressed in the present manuscript. Although the review for each section has been adequately addressed, several changes are required to update the manuscript.

Major:

#1. Conclusion: Because the meta-analysis was not performed comparing the above risk factors between men and women, the second sentence could not be drawn and endorsed in the present study, although the background and rationale behind COPD was well described in the text.

Minor:

#1. Abstract: Abbreviation of “CI” should be explained in the text.

#2. Methods/Study selection: “overweight” may be changed to “underweight” in the sentence.

Reviewer #3: The authors did a meta-analysis of published studies in COPD risk factors that focused in the populations of China. Out of the 2,449 studies found from January 2000 to December 2020, 17 studies matched their selection criteria. The statistical analysis of the COPD risk factors of these studies identified i) 2.5um particle exposure, ii) smoking history, iii) BMI, biomass burning emissions, and iv) family history of respiratory diseases as COPD risk factors for the people that reside in China. I believe that this kind of analysis is important and could help prevent COPD. However, grammar needs to be cleaned up and I do have some questions and a few recommendations for the authors:

Abstract

I understand the economic aspect of the disease, but the biggest impact of COPD is not the economic burden; it’s the impact in the quality of life of the COPD patients. They become prone to viral/bacterial/fungal infections which can worsen the already damaged lungs and could lead to death. I would suggest to include that in the abstract since this supports better the necessity/importance of your study.

The second phrase of the background section need rephrasing. The studies do to refer to Chinese population but to Chinese population that resides in China. Also, this study identified COPD risks factors that if taken under consideration could help with the early-identification and prevention COPD in a large part of this population.

At the methods section please replace the word “17 articles were included” with “selected”.

At the conclusion section I would include never starting smoking. I would be more specific regarding the weight, please include the BMI instead of the “reasonable weight”. The phrase “staying vigilant to changes in the health of a child’s respiratory tract” is confusing. I believe you are referring to respiratory track infections. Are you suggesting to take precautions so the child doesn’t get respiratory infections? Or treat these infections in a timely manner? What is the critical age for kids, up until what age they need to be protected?

Methods

Why was the search limited to English language? I would expect that including studies published in Chinese would enrich your data and provide a better insight to the whole scientific community that doesn’t understand Chinese thus cannot access those studies.

When you state all participants are from China, do you refer to Asian population that resides in China or any population of any race that resides in China?

When you state that case diagnosis is clear, what does this mean? What clinically/imaging confirmed means? Pulmonary function test? What % decline in FEV1/FVC/DLCO? Is imaging referring to CT scan?

Results

Figure 1/: I am not sure how politically correct is to include Taiwan as part of China (paper by TC Chan 2015).

How many years exposure to PM2.5 increases the risk for developing a COPD?

I would like more information regarding the age of the children that had respiratory infections, the type of the infection (viral/bacterial/fungal) and the severity of the infection. Also, were these children smoking or exposed to second hand smoke? Did these children have history of respiratory diseases? What was their pM2.5 exposure? How many times did they get infected? It is too vague to state that any respiratory infection during childhood could lead to COPD. Is there any evidence to narrow down this risk factor?

Regarding the family history of respiratory disease, could you please provide more information regarding the respiratory diseases involved? (Asthma, COPD, IPF, etc.) Also, I am assuming you are referring to chronic conditions.

I would also be interested to see if any of these studies identified any genetic markers apart from environmental factors.

Discussion

I would like to see a section that clearly states that this study provides evidence, which could help advance the medical field. What is the innovation, new knowledge gained?

Furthermore, how could this evidence pass into clinical practice in China?

6. PLOS authors have the option to publish the peer review history of their article (what does this mean?). If published, this will include your full peer review and any attached files.

Reviewer #1: No

Reviewer #2: No

Reviewer #3: **Yes: **Efthymia Iliana Matthaiou, PhD

---

## [Author Response · Author response to Decision Letter 0]

3 Dec 2021

RESPONSE TO REVIEWERS

-Reviewer #1

In this study, the authors explored epidemiological evidence regarding risk factors for chronic obstructive pulmonary disease in the Chinese population using systematic review and meta-analytic methods. Although the titles and the research question were interesting, several serious methodologic concerns must be addressed before being re-considered for publication. I suggest that this manuscript undergo a major revision.

1. Introduction: The structure of the introduction section should be re-organized. It is still not clear to me why this meta-analysis should be conducted. More detail on the study rationale should be added. Suppose the authors were to conduct a study to explore COPD risk factors specifically for the Chinese population. In that case, there should be a solid theoretical background of why there is a good reason to believe that COPD risk factors in the general population should be different from the Chinese population.

Response: Thank you for your remarks. Although COPD risk factors in the general population should be similar among the Chinese population, there are some specific factors special in China, including climate change, environmental pollution, public health literacy, and medical technology. Therefore, we believe it is reasonable to conduct this study to provide comprehensive evidence regarding risk factors for COPD in the Chinese population. Narrative related to theoretical background has been added. Please see detailed revisions on p.3, lines 57-61.

2. Methods

 (1) It is questionable whether restricting comprehensive literature searches to only English biomedical databases was adequate for a rather specific question for a specific population. Shouldn’t large, high-quality studies reported in Chinese be included?

Response: With our apologies, the practice of restricting comprehensive literature searches to only English biomedical databases is not adequate enough, therefore, we re-screened the publications and included the large-scale high-quality studies reported in Chinese. Please see the detailed revisions on p. 4, line 108.

(2) A searching strategy (i.e., keywords used, number of records identified for each keyword) for each database should be provided in a supplementary appendix.

Response: Thank you. The searching strategy includes including keywords used has been added. Please see the “S1 Table. Search strategy” in the Supplementary appendix on p.1, lines 3-34.

(3) The study selection criteria were unclear and subjective. Please clarify all of the criteria and make it more objective. For example, how do you define a strictly controlled research method?, how do you define when the study population is clearly defined?, how do you define when the case diagnosis is clear? Etc.

Response: Thank you. We defined COPD patients as subjects with FEV1 / FVC less than 70% after using post-bronchodilator, or patients diagnosed with chronic bronchitis, emphysema or other diseases dominated by airflow restriction by doctors. A clear definition has been added. Please see detailed revisions on p. 4, lines 120-122.

(4) Please change the terms “relative risk” to “risk ratio”.

Response: With our apologies, the terms has been corrected. Please see the revision on p. 5, line 123.

(5）“Literature review type articles” should not be stated in the exclusion criteria as the inclusion criteria had already stated that only published case-control, cohort, or cross-sectional studies would be included.

Response: Thank you. The exclusion criteria have been corrected. Please see detailed revisions on p. 4, lines 124-128.

(6) I am not quite sure that research with inaccurate design methods, low reliability, or poor quality should be excluded. I believe the idea of performing a systematic review was to comprehensive include all relevant studies possible. Exclusion of studies with poor methodological design might give rise to publication bias. Including these studies within this systematic review and performing a subgroup or sensitivity analyses to exclude these problematic studies might be a good alternative way.

Response: Considering that excluding studies with poorly designed methods may lead to publication bias, we re-included these studies and performed sensitivity analysis to exclude these studies. Please see the specific steps and results on p.5, lines 144-153.

(7)Details on who perform the screening, searching, and selection of records should be included within the methods section.

Response: The information related to filtering, searching and selecting records has been added. Please see detailed revisions on p. 4, lines 126, 130 and 141.

(8) In the data extraction section, “other basic information” should be further clarified. Also, within this section, potential factors that the authors intended to explore should be pre-specified.

Response: With our apologies that the expression "other basic information" is not rigorous. We deleted the “other basic information” when revising the article, and replaced it with a complete list of all extracted data for clearer expression. The complete list includes (I) the first author; (II) year of publication; (III) the area of the research; (IV) sample size; (v) type of research method; (vi) OR/RR value for potential risk factors for COPD and 95%CI provided by the study. Please see detailed revisions on p. 5, lines 130-133.

(9) In the statistical analyses section, pooling methods should be specified. The authors should pre-specified in detail when to perform fixed effects pooling or random effects pooling.

Response: We used the random-effects model when heterogeneity across the studies was large (I250%, P0.05) and fixed-effects meta-analysis at small heterogeneity (I250%, P0.05). Please see detailed revisions on p. 5, lines 147-150.

(10) Concerning I-squared statistics, the authors should specify the cut-off used for significant heterogeneity (and reference should be provided).

Response: Thank you. According to previous literatures, the cut-offs used for low, moderate, and high heterogeneity were 25%, 50%, and 75%, respectively. The random-effects model was used when heterogeneity across the studies was large (I250%, P0.05), while the fixed-effects meta-analysis at small heterogeneity (I250%, P0.05) was employed. The detailed statements and the related reference have been added on p. 5, lines 147-150.

(11)Again, I’m still not convinced that only studies with moderate-to-high quality should be included in this meta-analysis. I believe that this would lead to publication bias.

Response: Thank you. We have re-screened articles, included more studies, and re-conducted the meta-analysis. Please see detailed revisions in Table 1.

Results

(1) The authors stated in the results section that “257 papers were eliminated due to incomplete data, no adjustment for confounding factors, or results that were not significant.” However, these elimination criteria were not specified within the methods section.

Response: Apologies. The exclusion criteria have been implemented in the Methods section. Please see detailed revisions on p. 4, lines 124-128 and Fig 1.

(2)Again, I’m not convinced that excluding studies with insignificant results is a good approach for conducting systematic review.

Response: More studies were included and further sensitivity analyses were conducted. Please see detailed revisions on p. 4, lines 124-128.

(3) The sample size was reported as number of COPD cases, or the number of total people included? This should be clarified or separately reported.

Response: The sample size was reported as the total number of people included. Please see detailed revisions on p. 5, line 160.

(4) It is not clear how the authors arrange the sequence of the included studies on Table 1. I suggest that the authors stratify this table by study design and arrange the studies within each category by year of publication.

Response: We stratified Table 1 by study design and arrange the studies within each category by year of publication.

(5) Details on how the authors define considerable heterogeneity should be stated in the methods section, not in the results section.

Response: Thank you. We have explained the definition of considerable heterogeneity in the Methods section. Please see detailed revisions on p. 5, lines 147-150..

(6) Table 2, Table 3, and Figure 2 should be combined.

Response: Table 2, Table 3, and Figure 2 have been combined. We integrated the effect values and heterogeneity of the included studies into Table 2. Please see the detailed revisions in Table 2. 

(7) In Figure 2, not all significant predictors were included. It was unclear why the authors chose only some predictors to be presented in this figure.

Response: Thank you. The results of Figure 2 have been merged into Table 2. Please see detailed revisions in Table 2. 

(8) In Table 2 and Table 4, “=” should be changed to “≥”.

Response: With our apologies, the symbol has been corrected. Please see the detailed revisions in Table 2 and S2 Table. （We submit Table 4 as S2 Table in supplementary material ）

(9) Table 4 should be included as supplementary material.

Response: Table 4 has been submitted as supplementary material “S1 File. Quality evaluation of included studies”. 

(10) In the results section, the authors did not provide any numerical, tables, figures for subgroup analyses. Overall, the results section should be re-written. Essential data should be presented.

Response: Thank you. The Results section has been re-written and essential data have been provided. Please see detailed revisions on p. 8, lines 214-218. Essential data has been presented as S1 Data in supplementary material.

Conclusions

• The study conclusion is wrong. This meta-analysis did not show that reducing the exposure to risk factors would prevent or reduce the incidence of COPD. This meta-analysis only explores potential risk factors for COPD in the Chinese population.

Response: Apologies for the inaccurate statements. The conclusion has been corrected accordingly. Our meta-analysis explored potential risk factors for COPD in the Chinese population to provide some evidence for the early identification and prevention of high-risk COPD. Please see detailed revisions on p. 10, lines 326-334.

Figures

• PRISMA flow should be updated to the 2020 version.

• The quality of all figures should be improved.

Response: Thank you. PRISMA flow has been updated to 2020 version, and the quality of all figures has been improved. Please see “S1 Checklist. PRISMA 2020 checklist” in the Supplementary appendix.

-Reviewer #2

The authors demonstrated the risk factors of COPD performing the systematic review and meta-analysis in Chinese population. Their article is likely to help readers to learn this field. According to their results, PM2.5, smoking history, BMI18.5kg/m2, exposure to biomass burning emissions, and family history of respiratory diseases were the risk factors. These findings let readers reconsider this field. Despite no description of new insights in this field, the review for each section has been adequately addressed in the present manuscript. Although the review for each section has been adequately addressed, several changes are required to update the manuscript.

Major:

#1. Conclusion: Because the meta-analysis was not performed comparing the above risk factors between men and women, the second sentence could not be drawn and endorsed in the present study, although the background and rationale behind COPD was well described in the text.

Response: Thank you for your remarks. Combined with the other reviewer’s comments, we included more studies and conducted further analyses. The results showed that male sex may be a potential risk factor for COPD. Therefore, gender is discussed and analyzed in the Discussion section. Please see detailed revisions on p. 9, lines 264-275.

Minor:

#1. Abstract: Abbreviation of “CI” should be explained in the text.

#2. Methods/Study selection: “overweight” may be changed to “underweight” in the sentence.

Response: Thank you. Abbreviation of “CI” has been explained on p.4, line 124. “Overweight” has been changed to “underweight” on p. 4, line 119. 

-Reviewer #3

The authors did a meta-analysis of published studies in COPD risk factors that focused in the populations of China. Out of the 2,449 studies found from January 2000 to December 2020, 17 studies matched their selection criteria. The statistical analysis of the COPD risk factors of these studies identified i) 2.5um particle exposure, ii) smoking history, iii) BMI, biomass burning emissions, and iv) family history of respiratory diseases as COPD risk factors for the people that reside in China. I believe that this kind of analysis is important and could help prevent COPD. However, grammar needs to be cleaned up and I do have some questions and a few recommendations for the authors:

Response: Thank you. We have double-checked the whole manuscript and cleaned up all the grammar errors. Please see the revised manuscript. 

Abstract

I understand the economic aspect of the disease, but the biggest impact of COPD is not the economic burden; it’s the impact in the quality of life of the COPD patients. They become prone to viral/bacterial/fungal infections which can worsen the already damaged lungs and could lead to death. I would suggest to include that in the abstract since this supports better the necessity/importance of your study.

Response: Thank you for your remarks. We do agree and the impact of COPD on patients' quality of life, which has been described in the Abstract section. Please see detailed revisions on p. 2, lines 21-22.

The second phrase of the background section need rephrasing. The studies do to refer to Chinese population but to Chinese population that resides in China. Also, this study identified COPD risks factors that if taken under consideration could help with the early-identification and prevention COPD in a large part of this population.

Response: Thank you. The second phrase of the Background section has been rephrased, and combined with the other reviewer’s suggestion, the definition of research object has been emphasized in this study. Please see detailed revisions on p. 2, line 24.

At the methods section please replace the word “17 articles were included” with “selected”.

Response: Thank you. The term has been corrected. Please see detailed revisions on p. 5, line 160.

At the conclusion section I would include never starting smoking. I would be more specific regarding the weight, please include the BMI instead of the “reasonable weight”. The phrase “staying vigilant to changes in the health of a child’s respiratory tract” is confusing. I believe you are referring to respiratory track infections. Are you suggesting to take precautions so the child doesn’t get respiratory infections? Or treat these infections in a timely manner? What is the critical age for kids, up until what age they need to be protected?

Response: Thank you. Based on your valuable suggestions, conclusions on smoking and weight control have been revised. In terms of childhood respiratory track infections, lung growth and development is related to pregnancy, birth, childhood, and adolescent exposure, therefore, any factors affecting lung growth and development during this period may increase the risk of COPD. As a result, we recommend taking interventions to protect children from respiratory infections or adopting active treatments to children with respiratory tract infection for early prevention of COPD. In addition, because the lungs of children aged 0 to 18 are immature and still undergoing growth, we believe that more attention should be paid to the pulmonary infection among children aged 0-18. Please see the detailed revisions on p. 9, lines 289-293.

First, many epidemiological studies have consistently shown that the probability of respiratory symptoms, decline of pulmonary function, prevalence and mortality of COPD in smokers are significantly higher than those in non-smokers. Barbara’ meta-analysis confirmed and quantified the causal relationship between COPD and smoking. The prevalence of COPD in smokers was higher than that in ex-smokers, and prevalence of COPD in ex-smokers was higher than that in never-smokers. Therefore, we advocated that people do not smoke, especially for never-smokers. Second, previous studies suggest that lower BMI (18.5 kg/m2) and higher BMI (BMI≥28 kg/m2) may be potential independent risk factors for COPD, so maintaining a reasonable weight (28＞BMI≥18.5 kg/m2) is a measure that cannot be ignored. 

Methods

Why was the search limited to English language? I would expect that including studies published in Chinese would enrich your data and provide a better insight to the whole scientific community that doesn’t understand Chinese thus cannot access those studies.

Response: Apologies. Restricting comprehensive literature searches to only English biomedical databases is not adequate enough, therefore, we re-screened the publications and included the large-scale high-quality studies reported in Chinese. Please see detailed revisions on p. 4, line 108.

When you state all participants are from China, do you refer to Asian population that resides in China or any population of any race that resides in China?

Response: The object of our study refers to Asian population that always live in China. We conducted this study to explore the risk factors of COPD in Chinese population for the reason that there were specific characteristics of the development of COPD in Chinese population compared with other groups due to the impact of climate change, environmental pollution, public health literacy, and medical technology. Please see detailed revisions on p. 4, lines 117-118.

When you state that case diagnosis is clear, what does this mean? What clinically/imaging confirmed means? Pulmonary function test? What % decline in FEV1/FVC/DLCO? Is imaging referring to CT scan?

Response: We defined COPD patients as those with FEV1 / FVC less than 70% after using post-bronchodilator, or patients diagnosed with chronic bronchitis, emphysema or other diseases dominated by airflow restriction by doctors. A clear definition has been added. Please see detailed revisions on p. 4, lines 120-121.

Results

Figure 1/: I am not sure how politically correct is to include Taiwan as part of China (paper by TC Chan 2015).

Response: Thank you. Chinese population include population living in mainland China, Hong Kong, Macau, and Taiwan. We are pretty sure that Taiwan is indeed a part of China, which is also supported by many publications. The World Happiness Report 2021 released by the United Nations clearly points out the region, “Taiwan Province of China”. The Global Competitiveness Report 2019 released by the World Economic Forum (WEF) assessed the driving forces of productivity and long-term economic growth in 141 global economies clearly pointed out the region, “Taiwan, China”.

How many years exposure to PM2.5 increases the risk for developing a COPD?

Response: We tried to explore the relationship between long-term exposure to PM2.5 and disease, however, the specific year of exposure was not indicated in the included literatures, which was a limitation of the study. Please see detailed revisions on p. 10, lines 316-318.

I would like more information regarding the age of the children that had respiratory infections, the type of the infection (viral/bacterial/fungal) and the severity of the infection. Also, were these children smoking or exposed to second hand smoke? Did these children have history of respiratory diseases? What was their pM2.5 exposure? How many times did they get infected? It is too vague to state that any respiratory infection during childhood could lead to COPD. Is there any evidence to narrow down this risk factor?

Response: Thank you. We are regret that literature included in this study did not indicate the age of the children having respiratory infections, the type of the infection (viral/bacterial/fungal), and the severity of the infection. Moreover, it was unknown about whether these children were smoking or exposed to second-hand smoke, whether they had a history of respiratory diseases, what was their PM2.5 exposure, and how many times have they been infected. Specifically, most of the included studies used the frequency of cough before the age of 14 as the evaluation standard of respiratory diseases in children, however, the rest were not specially defined. In addition, only one included literature classified the severity of childhood infection into frequent cough (cumulative 3 months per year), sometimes cough (1-3 months per year), and rare cough (less than 1 month per year). We feel sorry that it is difficult to find evidence to narrow down this risk factor, which has been added as a limitation of this study. Please see detailed revisions on p. 10, lines 318-320.

Regarding the family history of respiratory disease, could you please provide more information regarding the respiratory diseases involved? (Asthma, COPD, IPF, etc.) Also, I am assuming you are referring to chronic conditions.

Response: Thank you. In our study, patients whose parents and / or siblings have one of chronic bronchitis, emphysema, COPD, and bronchial asthma were counted as those with family history. It has been further explained in the revised manuscript on p. 10, line 299-302.

I would also be interested to see if any of these studies identified any genetic markers apart from environmental factors.

Response: Thank you. Although many genetic markers have been found to be associated with COPD phenotype, no genetic markers were found in the included studies. We believe that the exploration of genetic markers and the relationship between susceptibility genes and COPD is an important research direction in the future. Please see detailed revisions on p. 10, line 298.

Discussion

I would like to see a section that clearly states that this study provides evidence, which could help advance the medical field. What is the innovation, new knowledge gained?

Furthermore, how could this evidence pass into clinical practice in China?

Response: Thank you. Our study found that factors related with smoking exposure, body weight, and respiratory infections were significant risk factors and potential preventive strategies for COPD, which brought innovative evidence for clinical and public health practice in China. Clinical or public health practice can be taken for the early prevention of COPD. Please see detailed revisions on ABSTRACT, p. 2, lines 44-48, and Conclusions, p.10, line 327-334.

---

## [Editor Report · Decision Letter 1]

9 Dec 2021

Epidemiological evidence relating risk factors to Chronic Obstructive Pulmonary Disease in China: A systematic review and meta-analysis.

PONE-D-21-16397R1

Dear Dr. Wenya Yu,

We’re pleased to inform you that your manuscript has been judged scientifically suitable for publication and will be formally accepted for publication once it meets all outstanding technical requirements.

Kind regards,

Surasak Saokaew, PharmD, PhD, BPHCP, FACP, FCPA

Academic Editor

PLOS ONE
---

## [Editor Report · Acceptance letter]

15 Dec 2021

PONE-D-21-16397R1 

Epidemiological evidence relating risk factors to Chronic Obstructive Pulmonary Disease in China: A systematic review and meta-analysis 

Dear Dr. Yu:

I'm pleased to inform you that your manuscript has been deemed suitable for publication in PLOS ONE. Congratulations! Your manuscript is now with our production department. 

Kind regards, 

on behalf of

Dr. Surasak Saokaew 

Academic Editor

PLOS ONE